# The Influence of Temperature in the Al 2024-T3 Aluminum Plates Subjected to Impact: Experimental and Numerical Approaches

**DOI:** 10.3390/ma14154268

**Published:** 2021-07-30

**Authors:** Maciej Klosak, Rafael Santiago, Tomasz Jankowiak, Amine Bendarma, Alexis Rusinek, Slim Bahi

**Affiliations:** 1Laboratory for Sustainable Innovation and Applied Research, Universiapolis, Technical University of Agadir, Agadir 80000, Morocco; b.amine@e-polytechnique.ma; 2Advanced Materials Research Centre, Technology Innovation Institute, Abu Dhabi 51133, United Arab Emirates; rafael.santiago@tii.ae; 3Faculty of Civil and Transport Engineering, Poznan University of Technology, 60-965 Poznan, Poland; tomasz.jankowiak@put.poznan.pl; 4Laboratory of Microstructure Studies and Mechanics of Materials LEM3, University of Lorraine, 57-070 Metz, France; alexis.rusinek@univ-lorraine.fr (A.R.); mohamed-slim.bahi@univ-lorraine.fr (S.B.)

**Keywords:** Al 2024-T3 alloy, gas gun, numerical simulations, high-velocity impact

## Abstract

In this paper, perforation experiments were carried out and numerically modelled in order to analyze the response of 2024-T3 aluminum alloy plates under different initial temperatures T_0_. This alloy has a particular relevance since it is widely used as a structural component in aircrafts, but it is also interesting for other sectors of industry. A gas gun projectile launcher was used to perform impacts within initial velocities V_0_ from 40 m/s to 120 m/s and at temperatures varying from 293 K to 573 K. A temperature softening of the material was observed which was manifested in the reduction in the ballistic limit by 10% within the temperature range studied. Changes in the material failure mode were also observed at different test conditions. Additionally, a finite element model was developed to predict the material response at high velocities and to confirm the temperature softening that was observed experimentally. An optimization of the failure criterion resulted in a reliable model for such mild aluminum alloys. The results reported here may be used for different applications in the automotive and military sectors.

## 1. Introduction

The use of aluminum alloys is well reported in different industrial applications, mainly in the automotive and military domains. Due to its high strength and fatigue resistance, Al 2024 alloys are widely used in aircraft, especially in the wing, main body structures, and wall structures in the cargo hold, which are usually exposed to tension. It was in the late 1940s when industry became interested in composite materials called fiber-metal laminates (FML), in which aluminum plays a key role. FMLs offer attractive properties under impact loading conditions (explosion, impact); therefore, they are suitable for use in the aircraft industry. The aluminum alloy under study can also be found in other hybrid composites such S2-glass/epoxy composites [1]. Most of the structures that use this material can be subjected to high-velocity impacts, such as ballistic, bird-strikes, or explosions within a wide range of temperatures. This is why the study of the material impact response at different temperatures is required for designing and evaluation purposes.

Many authors have contributed to studies of the FMLs. Vlot [2] demonstrated outstanding performances of one commercial FML under quasi-static and dynamic loadings, dealing with 7075-T6 and 2024-T3 aluminum alloys. It is to be noted that despite the attractive impact performances, many FMLs have drawbacks associated with their relatively long processing cycles and interlaminar fracture toughness [3]. This imperfection was analyzed in [4] where many configurations of FMLs and termed TFMLs were developed. Interesting FML configurations were also studied in [5,6] the authors of which focused on 2024-T3 aluminum, which is the subject of the present study.

The alloy in question gained interest and it was successfully applied in an FML named GLARE [7] and patented in 1987. GLARE is lighter than the Al 2024 alloys by 10–15% and has a better performance in terms of yield stress and bearing capacity. It has similar ballistic properties [8]. A big commercial success of the composite was the double-deck Airbus A380 in which almost 500 m^2^ of its surface was made of GLARE.

Currently, numerical models have been widely used in industrial and academic studies for predicting the material and structure behaviors. However, the use of numerical models and a proper material characterization for extreme conditions, such as impact and perforation, still remain the subject of intensive research. Interesting studies were reported by Vlot, Santiago et al., Borvik et al., Gupta et al., Clausen et al., and Bendarma et al. [2,5,9,10,11,12,13,14] which are related to aluminum alloys under dynamic behavior using experiments and compared to numerical results.

In the present study, the perforation of aluminum panels impacted by a conical projectile is studied experimentally and numerically. The experimental program was proposed in order to identify the material ballistic limits and the residual velocities at different temperatures. The finite element model was also developed, and the numerical results were compared to experimental data. It was observed that the material response is influenced by the temperature: the ballistic limits and failure mode depended on temperature.

## 2. Experimental Approach

Dynamic perforation tests are rarely coupled with thermal analysis since gas guns are not frequently equipped with a thermal chamber. The usual approach to take the temperature into consideration in perforation tests is to carry them out at room temperature and to extrapolate the results using numerical simulations at high temperatures knowing the constitutive relation. Many authors have dealt with perforation analysis from theoretical approaches, as discussed by the authors in [15,16,17], to more practical considerations as reported by the authors in [12,18,19,20,21,22]. The thermal softening of the material is usually tested using quasi-static experiments and its extrapolation to high strain rates is often a rough simplification. In the present study, a gas gun with a thermal chamber provided the solution to overcome these limitations. More results using this new heating system have already been provided for different materials including metals (along with aluminum) or polymers [23,24,25,26,27,28]. Negative temperatures have also been investigated using a cooling system [29].

The gas-gun project launcher is presented in Figure 1 and is also described in [24]. The projectile is placed in the barrel (C) and then accelerated by the compressed air in the reservoir (A) released by a fast valve (B), in order to reach the expected impact velocity $V_{0}$. Then, the projectile impacts a plate with partial or complete perforation depending on the quantity of the kinetic energy transferred to the tested plate. A pair of laser sensors are used to measure the initial impact velocity $V_{0}$ (D) and a laser barrier is fixed behind the plate to measure the residual velocities $V_{R}$ (F). The apparatus contains a thermal chamber (E) in which a specimen (I) is fixed that allows a uniform temperature distribution to both sides of the specimen. A PID controller (H) is also used for maintaining a constant temperature.

The projectile used in the experiments had a ϕ_p_ 72° conical edge, and was 11.5 mm in diameter and 35 mm long, as can be seen in Figure 2a. A typical failure mode for metals in the form of petalling was analytically described by Atkins et al. [30] and studied by several authors, for example by Kpenyigba et al. [31]. In the present study, the petalling may have varied as a function of the temperature of the impact velocity but was not based on the projectile geometry. 

The panels used were 130 mm × 130 mm (the active part 100 mm × 100 mm) aluminum 2024-T3 flat plates, 1.0 mm thick, fully clamped along their perimeter with two rigid metal rigs, placed on both sides of the plates, as seen in Figure 2b.

The experiment was followed by efficient numerical simulations. For that purpose, the finite element model was developed using Abaqus/Explicit commercial software [32].

## 3. Numerical Modelling

The aluminum plate was simulated by a discretized model combined of C3D8R elements at the central 60 mm diameter region and C3D8I elements in the peripheral area, as can be seen in Figure 3. The projectile was modeled as a combination of C3D8R and C3D10 elements. A mesh sensitivity analysis was conducted in order to identify a suitable combination of panel response and processing time. The projectile was defined as a rigid body since the material it is made of is maraging steel with a yield stress of 2 GPa. The specimen was fixed alongside the entire perimeter assuming a complete clamping as during experimental tests.

The characteristics of the model size are as follows:
specimen: fine mesh in the middle with 226805 nodes, and 180096 elements C3D8R (four elements along the thickness, 0.2 mm × 0.2 mm × 0.3 mm); the remaining part has 10,540 nodes, and 7888 elements C3D8I (4 elements along the thickness, 2 mm × 2 mm × 0.3 mm; both parts of the specimen were tied in the analysis, the refined mesh part has a form of a circle of 6 cm in diameter in the region of contact between the two acting bodies;projectile: 9302 nodes including cylinder 960 C3D8R elements and cone 5516 C3D10M (tetra) elements (average size 1.5 mm); both parts of the projectile were tied in the analysis.

The Johnson–Cook constitutive relation was used to take into account experimental observations [33]. This thermo-viscoplastic material model described by Equation (1) determines the strain and strain rate hardening and the thermal softening of the material. In Equation (1), A is the yield stress, B and n are the strain hardening coefficients, C is the strain rate sensitivity coefficient, ${\dot{\varepsilon }}_{0}$ is the strain rate reference value and m is the temperature sensitivity parameter. The last bracket in Equation (1) describes the thermal softening of the material and reduces the limit of the Huber–von Mises equivalent stress $\bar{\sigma }$ from the reference value at temperature $T_{0}$ to zero at the melting temperature $T_{m}$. In Figure 4, the behavior of the 2024-T3 aluminum alloy (stress–strain curve) is presented based on works by Santiago [34] and Buyuk et al. [35] for two temperature values and compared with the Johnson–Cook model [33].
(1)$$\bar{\sigma }\left({\bar{\varepsilon }}_{\mathrm{pl}},{\dot{\bar{\varepsilon }}}_{\mathrm{pl}},T\right)=\left(A+{B\bar{\varepsilon }}_{\mathrm{pl}}{}^{n}\right)\left(1+\mathrm{Cln}\frac{{\dot{\bar{\varepsilon }}}_{\mathrm{pl}}}{{\dot{\varepsilon }}_{0}}\right)\left[1-{\left(\frac{T-T_{0}}{T_{m}-T_{0}}\right)}^{m}\right],$$

The failure criterion is often crucial to efficiently simulate the dynamic response of a material—this was discussed by Teng and Wierzbicki [36]. After an analysis of several approaches, the progressive damage and failure model proposed by Johnson and Cook was selected to simulate the complex behavior of the aluminum alloy [37,38] and to predict the initiation of damage in the material, taking into account the appearance of the strain softening leading to the material’s failure. The initiation criterion presented in Equation (2) has five independent parameters controlling the evolution of the equivalent plastic strain at failure initiation depending on the stress triaxiality, strain rate ${\dot{\bar{\varepsilon }}}_{\mathrm{pl}}$, and temperature T. The parameters for our values and previous approaches are provided in Table 1. The effect of two temperatures (293 K and 573 K) on the initial damage strain is presented in Figure 5. The constant criterion (${\bar{\varepsilon }}_{\mathrm{pl}}^{D}=0.3,\;0.7$—horizontal lines) and the one described by Johnson and Cook [33] are temperature insensitive. However, as the additional parameter $d_{5}$ in Equation (2) was calculated based on the current numerical simulations, the initiation damage criterion (black lines) is additionally given for two different temperatures. It should be noted that the forms of Equations (1) and (2) do not depend on the shape and size of the specimen.
(2)$${\bar{\varepsilon }}_{\mathrm{pl}}^{D}=\left[d_{1}+d_{2}\exp\left(-d_{3}h\right)\right]\left[1+d_{4}\ln\frac{{\dot{\bar{\varepsilon }}}_{\mathrm{pl}}}{{\dot{\varepsilon }}_{0}}\right]\left[1+d_{5}\left(\frac{T-T_{0}}{T_{m}-T_{0}}\right)\right]$$

After the damage initiation point defined by Equation (2), the material damage evolves, and the stress–strain curve descends. The damage response depends on the finite element dimensions; therefore, the mesh dependency of the results is minimized. The strain localization is due to softening results in dissipated energy decrease for the refined mesh. Finally, the fracture energy $G_{f}$ in Equation (3) is used to reduce mesh dependency. In this approach, the softening response after damage initiation is characterized by a stress-displacement $\bar{\sigma }-{\bar{u}}_{\mathrm{pl}}^{F}$ response.
(3)$$G_{f}=\overset{{\bar{\varepsilon }}_{\mathrm{pl}}^{F}}{\underset{{\bar{\varepsilon }}_{\mathrm{pl}}^{D}}{\int }}L\bar{\sigma }d{\bar{\varepsilon }}_{\mathrm{pl}}=\overset{{\bar{u}}_{\mathrm{pl}}^{F}}{\underset{0}{\int }}\bar{\sigma }d{\bar{u}}_{\mathrm{pl}},$$

An introduction of the regularization of the strain softening problem requires a determination of the characteristic finite element length L in the equation $d{\bar{u}}_{\mathrm{pl}}=L{\bar{d\varepsilon }}_{\mathrm{pl}}$. In the current work, the linear displacement softening parameter was used and one additional calibrated parameter ${\bar{u}}_{\mathrm{pl}}^{F}$ was introduced to the model. ${\bar{u}}_{\mathrm{pl}}^{F}$ is the effective plastic displacement at failure and it is related to the time of damage initiation [31]. The dimensionless damage parameter D increases from the value of 0 at the damage initiation point, then it grows linearly to 1 at ${\bar{u}}_{\mathrm{pl}}^{F}$ corresponding to the state when the material is completely damaged. The failure of the element is realized by its deletion from the mesh. While the scalar damage increases, the stiffness degradation progresses as shown in Equation (4).
(4)$$\sigma =\left(1-D\right)\bar{\sigma },$$

The additional material parameter connected to the material strain softening is the effective plastic displacement at failure ${\bar{u}}_{\mathrm{pl}}^{F}$. The effect of the material softening as the regularisation of the strain softening has been positively evaluated in this work. Most of the material parameters were extracted from the author’s previous works [34] and from Buyuk et al. [35], as summarized in Table 1. The Young’s module and Poisson’s ratio were 73.1 GPa and 0.29, respectively [35]. In order to include the adiabatic heating effect during the impact, the Taylor–Quinney model was used, where the inelastic heat fraction parameter is equal to 0.9. Additionally, for the thermomechanical analysis, the specific heat, $C_{p}$ = 900 J/kgK, density and ρ = 2770 kg/m^3^ were considered. It was assumed that general contact should be considered, including interior contact surfaces engaged during the failure or erosion of the mesh, with the friction parameter set at 0.2 [39,40,41]. The initial material temperature was set according to the experimental test performed at the initial conditions. It varied between 293 K and 573 K. The parameters A, B, n, $d_{1}$, $d_{5}$ and $T_{m}$ and ${\bar{u}}_{\mathrm{pl}}^{F}$ were calibrated based on our own quasi-static tensile and perforation tests, as can be seen in Table 1.

The failure criterion parameters (d_1_, d_2_) were fitted from numerical simulations. Two main criteria guided towards the correct values: the shape of failure mode and the global value of residual velocity (V_R_). The optimal choice of the parameters is discussed later in the text.

## 4. Results and Discussion

Table 2 summarizes the experimental results. The values of the initial ($V_{0}$) and residual ($V_{R}$) impact velocities are given as well as the dissipated energy during the perforation process (E) calculated from the loss of the kinetic energy.

The ballistic limit evaluation has interested several authors and its numerical approaches are well described in [42,43,44]. The experimental ballistic curves present a relationship between the initial impact velocity $V_{0}$ and the residual impact velocity $V_{R}$ and point out a ballistic limit which is denoted by the intersection of the ballistic curve with the X-axis. The main task of this experimental–numerical analysis was to build up ballistic curves for the analyzed temperatures. As shown in Figure 6, there was a small temperature effect on the ballistic curves within the temperature range of 293–573 K. The ballistic limit estimated was probably higher than 50 m/s. It is hard to say precisely, but in room temperature for the initial projectile velocity equal to 49.5 m/s, the projectile did not perforate the plate. Other tests close to the ballistic limit were not considered.

The numerical simulations reproduced the experimental behavior of the specimens. The kinematic process of petals forming was similar to the experimental findings: the number of petals observed varied between 6 and 8, which is shown in Figure 7 and Figure 8. The figures show the plastic equivalent strain calculated for both the damage initiation criteria (our own data) from Table 1. The distribution of the equivalent plastic strains (PEEQ) is slightly different in both cases. In our own work, the results implemented into the Johnson–Cook failure criterion, as shown in Figure 7b and Figure 8b, gave higher values because the material was in the compression zone (negative triaxiality has a higher strain at the initiation of damage as seen in Figure 5). In both cases the failure pattern is similar to the one obtained experimentally, although the Johnson–Cook failure criterion allowed a better pattern to be obtained. There was a characteristic bent of the petals as can be clearly seen in the photos. It is evident that the petals are bent. The simulations allowed the temperature increase in the petal tips to be measured: the maxima were approximately 85 K for the initial temperature $T_{0}$ = 293 K and about 62 K for the initial temperature $T_{0}$ = 573 K, respectively. The average strain rates in the crack zone were between 2500 l/s and 7500 l/s.

Figure 9 recapitulates numerical results for different failure criteria which are compared to the curves determined from the experiment.

## 5. Conclusions

The behavior of the aluminum 2004-T3 alloy under impact was analyzed experimentally and numerically. The principal aim of the study was to confirm the influence of temperature on the high velocity response of the aluminum panels under perforation experiments. The recorded experimental data helped the failure mode to be observed in the form of petals and allowed an effective constitutive law and an FE model for numerical simulations to be proposed. The phenomenological formula of Johnson–Cook was proposed to describe the mechanical material response and different failure criteria were considered. As a result, the progressive damage and failure model by Johnson and Cook gave the best results in terms of reproducing experimental ballistic properties. There was a significant unfavorable change in the petalling form when the simplified uniaxial failure criterion of maximum plastic strain at failure replaced the Johnson–Cook approach. The temperature dependence of the failure criterion was an important factor. The average number of petals reported in the tests and simulations was equal to 6–8. The maxima of temperature registered at the petals’ peaks in the numerical simulations was ∆T = 85 K and the average strain rate was of the order of 2500 l/s to 7500 l/s.

The ballistic limit for the conical projectile was equal to 48 m/s and did not change visibly within the analyzed temperature range; this result was observed in both the experiment and in the simulation. The comparison of the ballistic curves from the experiments and simulations exhibited a pronounced similarity.

In the next stages of the analysis schedule, the aluminum specimens will be analyzed experimentally to capture the adiabatic temperature increase development during perforation, and this will be performed using an infrared camera. On the other hand, the numerical model may be now extended to be applied in a FML (such as GLARE) analysis in which the Al 2024-T3 alloy is an important component. The reliable FE model will allow an extrapolation of the existing results to temperature and strain rate ranges which cannot be easily covered by an experiment.

## Figures and Tables

**Figure 1 materials-14-04268-f001:**
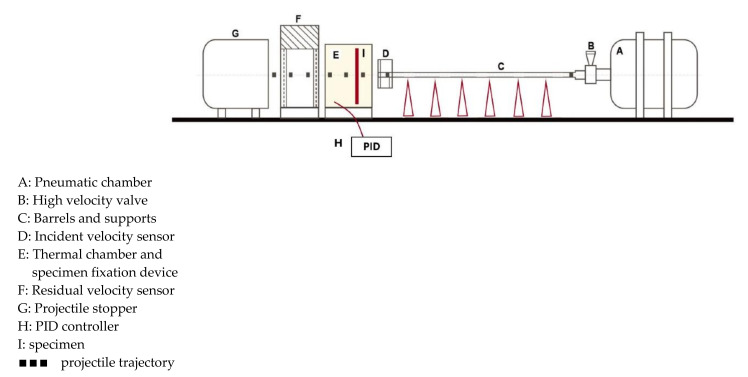
Scheme of gun set-up used for perforation tests at high impact velocities at different temperatures [24].

**Figure 2 materials-14-04268-f002:**
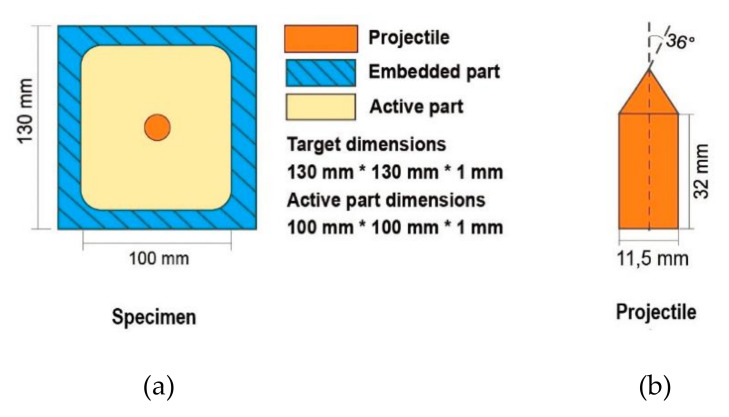
Details of (**a**) impacted panel and (**b**) projectile [24].

**Figure 3 materials-14-04268-f003:**
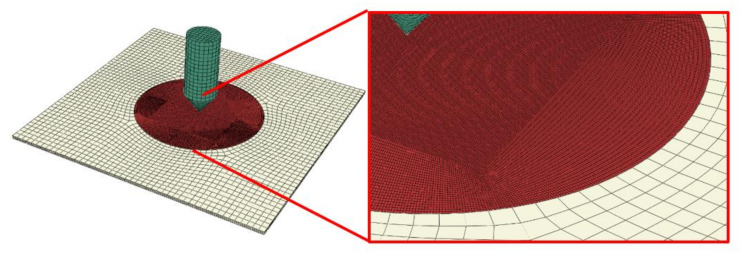
Finite element model used for modelling of the aluminium plate projectile.

**Figure 4 materials-14-04268-f004:**
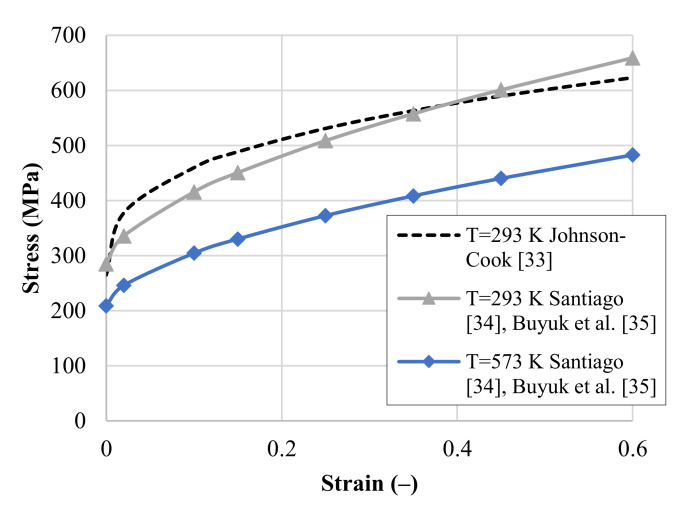
The calibrated material model (JC) based on experiments for the 2024-T3 aluminum alloy by Santiago [34], Buyuk et al. [35] for two different temperatures and strain rate of 1 l/s compared to experiments by Johnson and Cook [33]; (-) stands for a dimensionless parameter.

**Figure 5 materials-14-04268-f005:**
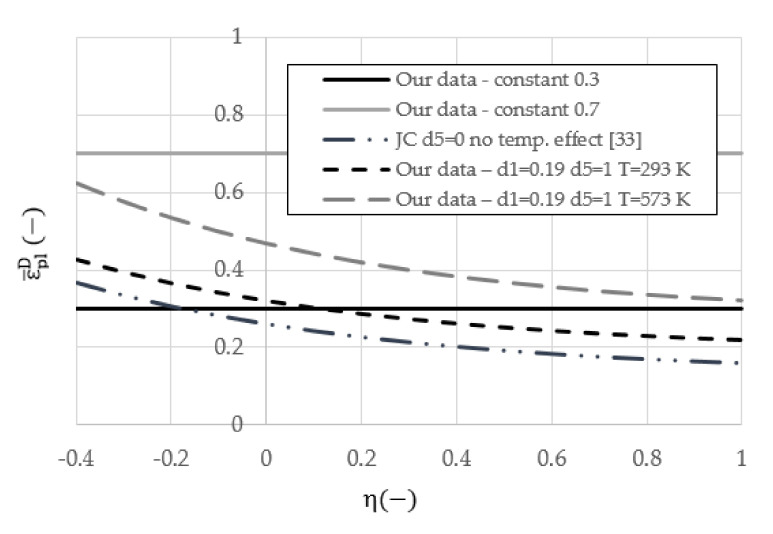
The considered initiation damage criteria for the 2024-T3 aluminum alloy as a function of triaxiality η for two different temperatures and strain rate 1.0 l/s; (-) stands for a dimensionless parameter.

**Figure 6 materials-14-04268-f006:**
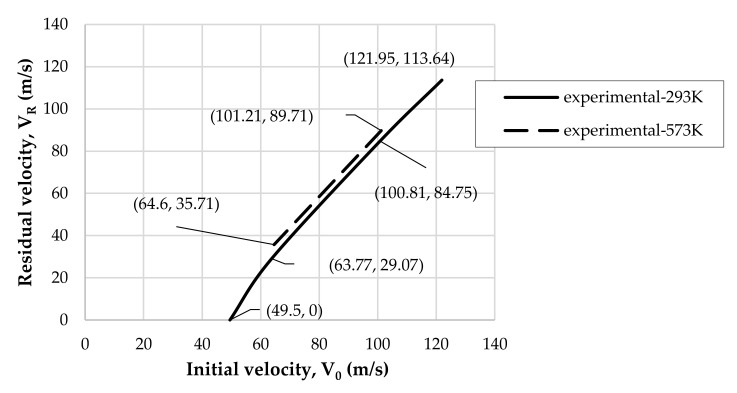
Ballistic curves $V_{0}$
vs. $V_{R}$ for experimental results: temperature range 293–573 K.

**Figure 7 materials-14-04268-f007:**
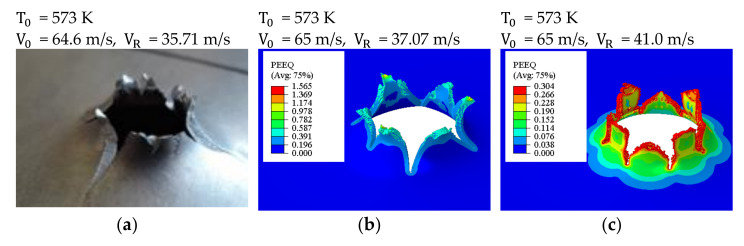
Petalling observed during the numerical simulation: $V_{0}$ = 65 m/s, $T_{0}$= 573 K; the sequence of pictures: (**a**) experiment, (**b**) simulation using the Johnson–Cook failure criterion (our own work), (**c**) simulation using a constant initiation damage parameter.

**Figure 8 materials-14-04268-f008:**
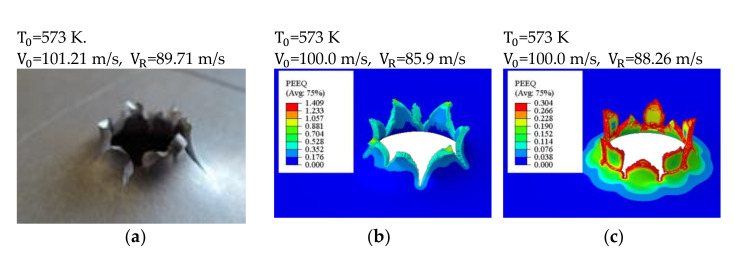
Petalling observed during the numerical simulation: $V_{0}$ = 100 m/s, $T_{0}$= 573 K; the sequence of pictures: (**a**) experiment, (**b**) simulation using the Johnson–Cook failure criterion (our own data), (**c**) simulation using a constant initiation damage parameter.

**Figure 9 materials-14-04268-f009:**
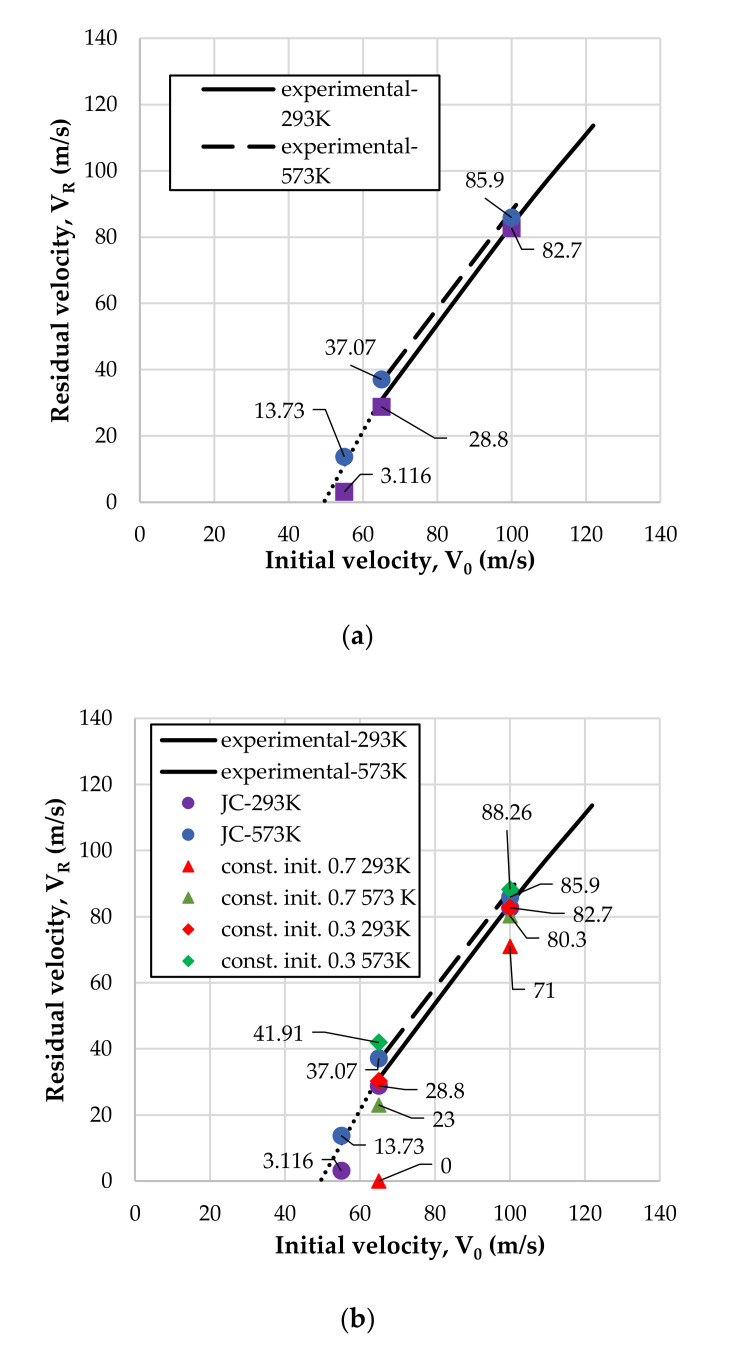
Ballistic curves $V_{R}$ vs
$V_{0}$: comparison of experimental and numerical results for $T_{0}$ = 293 K and $T_{0}$ = 573 K; (**a**) Johnson–Cook failure criterion (our work), (**b**) constant initiation damage parameter.

**Table 1 materials-14-04268-t001:** Material parameters for the Johnson–Cook model and the failure criterion.

**A (MPa)**	**B (MPa)**	**n (-)**	**C (-)**	**m (-)**	$T_{0}$ **(K)**	$T_{m}$ **(K)**	${\dot{\varepsilon }}_{0}$ **(l/s)**	
265	426	0.34	0.0083	1.7	293.15	775	1.0	Johnson and Cook [33]
284.9	504.81	0.5871	0.0083	1.7	293.15	900 *	1.0	Santiago [34], Buyuk et al. [35]
**d_1_ (-)**	**d_2_ (-)**	**d_3_ (-)**	**d_4_ (-)**	**d_5_ (-)**	${\bar{u}}_{pl}^{F}$ **(mm)**			
0.13	0.13	1.5	0.011	0	0.001			Johnson and Cook [33]
0.19	0.13	1.5	0.011	1	0.001			Johnson–Cook failure criterion—our work
0.3 or 0.7	0	0	0	0	0.001			Our work—constant ${\bar{\varepsilon }}_{\mathrm{pl}}^{D}$

N.B. the unit (-) describes dimensionless parameters. * our temperature data.

**Table 2 materials-14-04268-t002:** Summary of the perforation tests.

Test Ref.	Pressure (bar)	$V_{0}$ (m/s)	$V_{R}$ (m/s)	$T_{0}$ (K)	E (J)
AL08	1.3	49.5	0	293	10.1
AL02	2	63.77	29.07	293	13.28
AL01	5	100.81	84.75	293	12.28
AL09	7.6	121.95	113.64	293	8.07
AL04	2	64.94	35.21	473	12.27
AL03	5	100.8	86.21	473	11.24
AL07	2	64.6	35.71	573	11.94
AL06	2	64.1	n/a	573	n/a
AL05	5	101.21	89.71	573	9.36

## Data Availability

Data sharing is not applicable for this article.

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
