# Peer review of "The Influence of Temperature in the Al 2024-T3 Aluminum Plates Subjected to Impact: Experimental and Numerical Approaches"

_materials, 2021, doi:10.3390/ma14154268_

Round 1

Reviewer 1 Report

The paper is to investigate experimentally and simulate numerically some mechanical impact properties of 2024-T3 aluminum alloy plates under different temperatures. The alloy is widely used in aircrafts and other industries as a part of composite materials. These materials are subjects to impact and explosions, so the study is very important. Pure numerical model study of materials at extremal condition does not give a complete adequate description of all processes and results and experimental investigations are necessary for impact effect study.

 A significant difference from other experiments is the use of a thermostat by the authors to set and maintain the temperature of the sample under study. Two temperature values were selected for the experiments, 293 and 573 K. The sample was processed with a projectile using a gas-gun launcher. The authors described the experimental setup and conducted an experiment illustrating the possibilities of their method.

The numerical modelling followed the experiment. The simulation reflects a softening the alloy at higher temperature. The result for 293K is well consisted with a well-known Johnson -Cook phenomenological model. The authors included strain softening and plastic displacement  in their evaluation.

As result, the authors got ballistic curves for two different temperatures. Their experiment and numerical study show a small temperature effect on ballistic curves.

The authors demonstrated a good understanding of the literature of the field and cite important sources. The background knowledge is well illustrated. The physics and mathematics of the method is thoroughly described.

I recommend this article for publication.

Author Response

Dear Sir or Madam,

I do appreciate this positive review. Some minor improvements in the content have been still done during a reviewing process. I hope the paper will be accepted and published shortly.  

The corrections in the manuscript use the red colour for the text improvements/corrections and the blue one for English corrections.

We would like to thank the reviewers allowing to increase the quality of our paper.

Kind regards,

Maciek Klósak

Reviewer 2 Report

The paper describes an experimental and theoretical study of the strength of an aluminum alloy at various temperatures. The results of the work are potentially interesting, but they are not presented clearly enough. The following comments should be taken into account.

(1) All values included in formula (1) and subsequent formulas must be explained. It is unclear whether these formulas take into account the shape and size of the sample. If the shape is not taken into account, it is not clear why a rectangular model of the sample was used (round sample is much more suitable).

(2) The characters "(-)" are not clear; perhaps they should be replaced with "dimensionless".

(3) Table 1, which of these values are the fitting coefficients? What are the statistical errors of these values?

(4) Table 2, why is T0 measured in m / s? Table 2, it is not explained what E is. Line 187, the parenthesis is omitted.

(5) English should be significantly improved.

Author Response

Dear Sir or Madam,

My comments to your review are given in green in the attached file. The corrections in the manuscript use the red colour for the text improvements/corrections and the blue one for English language improvements.

Kind regards,

Maciek Klósak
